# Immunomorphological Pattern of Molecular Chaperones in Normal and Pathological Thyroid Tissues and Circulating Exosomes: Potential Use in Clinics

**DOI:** 10.3390/ijms20184496

**Published:** 2019-09-11

**Authors:** Celeste Caruso Bavisotto, Calogero Cipolla, Giuseppa Graceffa, Rosario Barone, Fabio Bucchieri, Donatella Bulone, Daniela Cabibi, Claudia Campanella, Antonella Marino Gammazza, Alessandro Pitruzzella, Rossana Porcasi, Pier Luigi San Biagio, Giovanni Tomasello, Everly Conway de Macario, Alberto J. L. Macario, Francesco Cappello, Francesca Rappa

**Affiliations:** 1Department of Biomedicine, Neuroscience and Advanced Diagnostics (BIND), Institute of Human Anatomy and Histology, University of Palermo, 90127 Palermo, Italy; celestebavisotto@gmail.com (C.C.B.); rosario.barone@unipa.it (R.B.); fabio.bucchieri@unipa.it (F.B.); claudia.campanella@unipa.it (C.C.); antonella.marino@hotmail.it (A.M.G.); alexpitruzzella@libero.it (A.P.); giovanni.tomasello@unipa.it (G.T.); francapp@hotmail.com (F.C.); 2Euro-Mediterranean Institute of Science and Technology (IEMEST), 90100 Palermo, Italy; econwaydemacario@som.umaryland.edu (E.C.d.M.); ajlmacario@som.umaryland.edu (A.J.L.M.); 3Institute of Biophysics, National Research Council, 90100 Palermo, Italy; donatella.bulone@pa.ibf.cnr.it (D.B.); pierluigi.sanbiagio@cnr.it (P.L.S.B.); 4Department of Surgical Oncology and Oral Sciences, University of Palermo, 90127 Palermo, Italy; calogero.cipolla@unipa.it (C.C.); giuseppa.graceffa@unipa.it (G.G.); 5Institute of Biomedicine and Molecular Immunology, National Research Council, 90100 Palermo, Italy; 6Department “G. D’Alessandro”, Pathology Institute, University of Palermo, 90127 Palermo, Italy; daniela.cabibi@unipa.it (D.C.); r.porcasi@libero.it (R.P.); 7Department of Microbiology and Immunology, School of Medicine, University of Maryland at Baltimore-Institute of Marine and Environmental Technology (IMET), Baltimore, MD 21202, USA

**Keywords:** thyroid gland, papillary carcinoma, molecular chaperones, heat shock proteins (Hsp), goiter, exosomes, diagnosis

## Abstract

The thyroid is a major component of the endocrine system and its pathology can cause serious diseases, e.g., papillary carcinoma (PC). However, the carcinogenic mechanisms are poorly understood and clinical useful biomarkers are scarce. Therefore, we determined if there are quantitative patterns of molecular chaperones in the tumor tissue and circulating exosomes that may be useful in diagnosis and provide clues on their participation in carcinogenesis. Hsp27, Hsp60, Hsp70, and Hsp90 were quantified by immunohistochemistry in PC, benign goiter (BG), and normal peritumoral tissue (PT). The same chaperones were assessed in plasma exosomes from PC and BG patients before and after ablative surgery, using Western blotting. Hsp27, Hsp60, and Hsp90 were increased in PC in comparison with PT and BG but no differences were found for Hsp70. Similarly, exosomal levels of Hsp27, Hsp60, and Hsp90 were higher in PC than in BG, and those in PC were higher before ablative surgery than after it. Hsp27, Hsp60, and Hsp90 show distinctive quantitative patterns in thyroid tissue and circulating exosomes in PC as compared with BG, suggesting some implication in the carcinogenesis of these chaperones and indicating their potential as biomarkers for clinical applications.

## 1. Introduction

The thyroid is one of the major components of the endocrine system and its pathological alterations in structure function can cause serious diseases. For example, papillary carcinoma (PC) is the most frequent malignancy not only of the thyroid but also of the entire endocrine system [1,2]. However, many aspects of tumorigenesis in the thyroid, for example the possible role of molecular chaperones in cancer initiation/progression and their potential for diagnosis and disease follow-up, are still poorly understood. We therefore focused on PC to determine if there are quantitative patterns of molecular chaperones in the tumor tissue that may be useful in diagnosis and that may provide clues on their participation in the carcinogenic process. Various molecular chaperones are heat shock proteins (Hsp) and here we will use the terms Hsp and chaperone as synonyms, following the literature usage. Typically, chaperones are involved in the maintenance of protein homeostasis and play a cytoprotective role against stress [3]. Some chaperones are implicated in carcinogenesis since they favor tumor cell growth, multiplication, and dissemination [4]. In view of this information, and considering the emerging role of exosomes in physiology and carcinogenesis and the fact that these microvesicles carry molecular chaperones [5,6,7,8,9], we extended our study to the characterization of the quantitative patterns of molecular chaperones in blood exosomes, presumably released by the tumor, in patients with PC. For a baseline to compare quantitative data from PC, we used the normal peritumoral thyroidal tissue from the PC patients, and thyroid from patients with non-toxic goiter (benign goiter or BG).

In thyroid tissues, by applying immunohistochemistry we not only quantified the immunopositivity revealed by the various anti-Hsp antibodies but we also characterized the morphology of the reaction and its distribution inside and outside cells. Thus, we used all the information provided by immunohistochemistry and describe the end result as “immunomorphology.” This term is a more comprehensive and better descriptor of the analyses carried out in our work than immunohistochemistry. The latter term in our usage describes the technique, while immunomorphology encompasses various pieces of information provided by immunohistochemistry. In this way, we studied Hsp27, Hsp60, Hsp70, and Hsp90, which have all been implicated in carcinogenesis [10,11,12,13,14,15,16,17,18]. In addition, we quantified these chaperones in circulating exosomes using Western blotting. We compared tissue data from patients with PC and BG, and plasma exosomes from PC patients before and after surgical tumor ablation. The aim of our study was to evaluate the tissue levels of Hsps by immunohistochemistry in a comparative analysis, and observe their localization in the two pathological thyroid conditions. This was necessary because there are very few pertinent data in the literature despite the facts that molecular chaperones are known to be involved in carcinogenesis and that exosomes carry molecular chaperones reflecting the contents of their cells of origin and, thereby, offer the opportunity to be used as biomarkers for diagnosis and disease follow-up.

## 2. Results

### 2.1. Quantitative Immunohistochemistry

The immunomorphological analysis revealed that the Hsp27 level in BG tissue was 30% while in PC it was 60%; the Hsp60 level was 5% in BG while in PC it was 92%; the Hsp70 level was less than 5% in BG and PC; and the Hsp90 level was 10% in BG and 80% in PC. Statistical analysis showed a significant (*p* ≤ 0.0001) increase in the levels of Hsp27, Hsp60, and Hsp90 in PC as compared with BG (Figure 1A,B). In contrast, no difference was found in the levels of Hsp70.

The chaperones were also evaluated in healthy peritumoral tissue (PT) of PC samples. In PT the average percentage of immunopositivity of Hsp27 was 25%, that of Hsp60 was 6%, that of Hsp70 was 5%, and that of Hsp90 was 8%. These results were similar to the immunopositivity percentages found in BG and were significantly lower than those pertaining to PC, except for the Hsp70 levels (Figure 1C). Furthermore, we performed an evaluation of the cellular localization of immunopositivity for Hsp27, Hsp60, and Hsp90. Hsp27 immunolocalized in the cytoplasm and in the perinuclear region in BG cells, while it was in the cytoplasm and in the plasma–cell membrane or close to it in PC cells (Figure 2).

Hsp60 immunopositivity was present in a very low percentage of BG cells as granules in the cytoplasm, but in PC many cells were positive with a diffuse pattern, with immunopositivity also present in the plasma–cell membrane and close to it. Hsp90 was immunolocalized in the cytoplasm of BG cells as well as in PC cells, but in the latter the immunopositivity was visible also in the plasma–cell membrane and close to it.

### 2.2. Quantitative Analysis of Exosomes from Plasma from PC Patients

Exosomes were purified from plasma from BG patients, and from plasma from PC patients before and after ablative surgery. The levels of the exosome markers Alix and CD81 were assessed in all exosomes by Western blotting (WB) (Figure 3A). Dynamic light scattering (DLS) quantitative analysis showed that the mean value of the number of plasmatic exosomes of patients with BG was 7.213 × 10^13^ ± 2.394 × 10^13^ and the mean value of sizes was 41.012 ± 7.739 nm. In PC patients, the number of plasmatic exosomes after surgery decreased from 3.294 × 10^13^ ± 1.339 × 10^13^ to 2.223 × 10^13^ ± 8.163 × 10^12^, but the difference, although appreciable, was not statistically significant. On the contrary, in PC the size of exosomes after surgery was significantly greater than that before it (Figure 3B). Transmission electron microscopy (TEM) and atomic force microscopy (AFM) images demonstrated the typical exosomal features in all our preparations (Figure 3C,D).

Hsp27, Hsp60, and Hsp90 were present in the exosomes, but at different levels in BG and PC, being significantly higher in PC. Moreover, in PC the levels of the three chaperones were higher before surgery than after it (Figure 4A–D).

In brief, the levels of Hsps in the exosomes of patients with PC before surgery were significantly higher than those in the exosomes from the same patients after surgery and those from BG patients.

## 3. Discussion

We determined the levels of Hsp27, Hsp60, Hsp70, and Hsp90 in thyroid papillary carcinoma (PC) and compared them with those in non-toxic goiter (simple or benign goiter, BG). The latter is a pathological condition in which the thyroid gland is generally enlarged (diffuse goiter) or presents localized nodules, usually several (multinodular goiter). Its etiology is complex, including genetic and environmental factors, and its incidence is closely related to a deficit in the iodine dietary intake. The microscopic morphology typically includes macro- and micro-follicles surrounded by follicular cells that are columnar or flattened.

PC is the most common thyroid malignant epithelial tumor that, in the classical form, shows true papillae surrounded by cells with typical malignant features.

Our results showed increased levels of Hsp27, Hsp60, and Hsp90 in PC in comparison with BG and normal peritumoral tissue. The cellular localization of these chaperones was different in the two groups studied. In BG cells, the chaperones were localized in physiological sites, while in PC cells they appeared differently—namely, they were more abundant in the cytoplasm, as well as localized in the plasma–cell membrane or close to it. These observations suggest that the chaperones may be secreted via extracellular vesicles, for instance. The cellular distribution of Hsps usually changes during the carcinogenic process [6,19,20,21]. Typically, they localize in the mitochondria (Hsp60) or in the cytoplasm of normal cells. Meanwhile, in cancer cells they accumulate in the cytoplasm and the plasma–cell membrane [6,14]. After their accumulation inside the cell, Hsps can be actively released into the extracellular space and in the blood [22]. These findings are in agreement with data pertaining to other types of human cancer. For example, our previous observations showed that the levels of Hsp60 and Hsp90 were higher in large bowel adenocarcinoma compared with normal mucosa [23], and that Hsp60 levels were higher in the pre-neoplastic and neoplastic lesions as compared with normal mucosa in the large bowel and in the uterine cervix [4,24]. Likewise, the higher tissue levels of Hsp27 we found in PC are in agreement with data published by other authors who have shown increased levels of Hsp27 in gastric and esophageal carcinoma, as well as an association of these increased levels with a poor prognosis [25,26].

We found no significant differences in the levels of Hsp70 between the two thyroid tissues, PC and BG; the levels were below 5% in both of them. This result is interesting since it contrasts observations by others suggesting an involvement of Hsp70 in carcinogenesis in other tissues, via participation in the mechanisms of apoptosis [27]. Hsp70 would interfere with apoptosis by binding to APAF-1, inhibiting its oligomerization, and blocking apoptosome assembly, thereby increasing the probability of cancer cell survival [4,28,29]. Along these lines, it should be mentioned here that mortalin, the mitochondrial version of Hsp70, has been found to be increased in PC and in another thyroid cancer, follicular carcinoma [30].

Our immunohistochemical study is, to our knowledge, the first in which a comparative evaluation of the levels of Hsp27, Hsp60, and Hsp90 was carried out in two types of thyroid disorders, one benign and close to normal morphological conditions, such as BG, and the other, PC, neoplastic and morphologically quite different from normal thyroid tissue. We carried out an immunomorphological analysis since we performed a quantitative assessment of the immunopositivity. Also, we observed the cellular localization of the Hsps studied. In view of our results, we decided to assess the Hsp27, Hsp60, and Hsp90 levels in circulating exosomes obtained from plasma. We did not evaluate Hsp70 in exosomes because the immunohistochemical experiments showed no difference between the two groups studied. DLS analyses of the plasmatic exosomes showed a decrease in their number after surgical resection of the tumor in PC. Also, an increase in exosome size was observed after tumor resection in patients with PC, but this trend, although clear, was not statistically significant. Consequently, it may be postulated that the size of plasmatic exosomes is a parameter deserving scrutiny in future research for assessing the value of these vesicles in discriminating cancer patients from healthy subjects [31].

In addition, this study shows for the first time the exosomal levels of Hsp27, Hsp60, and Hsp90 in the plasma of patients with a thyroid cancer, PC, before and after surgery, as compared with those of patients with the benign disease BG. These results seem promising for the consideration of exosomal Hsps, whose role in tumor progression in malignant thyroid diseases is not yet fully understood, as possible biomarkers useful in diagnosis, and for the assessment of prognosis and response to treatment with the advantage of requiring minimally invasive procedures. Our results are supported by previous observations on Hsp60 in exosomes and the pathophysiology of colon cancer [7]. Increasing evidence suggests that exosomes are an important component of the tumor microenvironment and may play a key role in tumor progression and metastasis dissemination (reviewed in [30]). Exosomes can deliver a range of molecules that participate in the modulation of the tumor microenvironment, for instance regulating gene expression in the target cells and the functioning of the immune system, thus generating a pro-metastatic niche [32,33,34,35,36]. The molecular content of exosomes depends on the status of parental cells and consists of proteins, lipids, and nucleic acids (DNA, non-coding RNA) [37,38,39]. Consequently, there is much interest in them as possible markers useful for diseases monitoring and as potential carriers of anticancer drugs [40,41,42,43]. Some evidence supports the hypothesis that the number and the shape of plasmatic exosomes are directly related to the tumor mass [44]. Hsps are part of the exosomes’ cargo [5] and some of them, e.g., Hsp60 and Hsp70, can mediate immunomodulatory effects and immune response [6,7,8,9].

## 4. Materials and Methods

### 4.1. Patients

The patients included in this study were undergoing thyroidectomy at the Department of Surgical, Oncological, and Oral Sciences at the University of Palermo, from April 2017 to May 2018.

The study was conducted in accordance with the Declaration of Helsinki, and the protocol was approved by the Ethics Committee of University Hospital AUOP Paolo Giaccone of Palermo (Approval Number: N° 05/2017 of 05/10/2017). All subjects signed their informed consent. Two blood samples from each subject were taken, the first one day prior to the surgical removal of the thyroid, and the second one week after, on the day of medication. These blood samples were used for exosomal isolation. All the thyroids, surgically removed, were sent to the Institute of Surgical Pathology for histological diagnosis. Eighteen patients (14 women and four men; mean age 60.1 ± 8.3 years) with diagnosis (clinical-laboratory-histological) of non-toxic (benign) goiter (BG) and 13 patients (11 women and two men; mean age 50.2 ± 11.5 years) with diagnosis of papillary carcinoma (PC) were included in this study (Appendix A).

The samples of thyroid tissue from BG and PC patients were obtained from the Institute of Surgical Pathology at the Department of Health Promotion, Mother and Child Care, Internal Medicine, and Medical Specialties at the University of Palermo. All tissue samples were used for immunohistochemistry.

### 4.2. Immunohistochemistry

Immunohistochemistry was performed on 5 micron thick sections of paraffin-embedded tissue. The sections were dewaxed in xylene for 30 min at 60 °C and rehydrated at 22 °C by sequential immersion in a graded series of alcohols. Subsequently, the sections were immersed for 8 min in sodium citrate buffer (pH 6) at 95 °C for antigen retrieval and, afterwards, immersed for 8 min in acetone at −20 °C to prevent the detachment of the sections from the slide. After washing the sections with PBS (phosphate buffered saline, pH 7.4) for 5 min at 22 °C, the detection of Hsp60, Hsp70, and Hsp90 was performed by the streptavidin–biotin complex method using a Histostain^®^-Plus Third Gen IHC Detection Kit (Life Technologies, Frederik, MD, USA; Cat. No. 85–9073). The immunohistochemistry procedure was performed as described previously [23]. The primary antibodies used were as follows: against human Hsp60 (rabbit anti-Hsp60 polyclonal antibody, Santa Cruz Biotechnology, Inc., Santa Cruz, CA, USA clone H-300, cat. N°: sc-13966, dilution 1:300); against human Hsp70 (Santa Cruz Biotechnology, Inc., clone W27, dilution 1:200); against human Hsp90 (Santa Cruz Biotechnology, Inc., clone F-8, dilution 1:200). Appropriate positive and negative (isotype) controls, were run concurrently. The reactions for Hsp27 were performed using an IHC goat kit (Cell and Tissue Staining Kit, R&D Systems, Inc., Minneapolis, MN, USA, Cat N° CTS008). After *deparaffinization,* the histological sections were treated for 5 min with Peroxidase Blocking Reagent (Cell and Tissue Staining Kit, R&D Systems, Inc.) to inhibit endogenous peroxidase activity and, after another wash with PBS for 5 min, with serum-blocking reagent D (Cell and Tissue Staining Kit, R&D Systems, Inc) for 15 min to block non-specific antigenic sites. Since the detection is based on the formation of the avidin–biotin complex, the sections were treated with avidin-blocking reagent (Cell and Tissue Staining Kit, R&D Systems, Inc) for 15 min (all previous steps were carried out at 22 °C). After washing with PBS, the sections were incubated with biotin-blocking reagent for 15 min (Cell and Tissue Staining Kit, R&D Systems, Inc.) and then incubated overnight at 4 °C with a primary antibody against human Hsp27 (goat anti-Hsp70 polyclonal antibody, Santa Cruz Biotechnology, Inc., dilution 1:150). The sections were then washed three times in PBS for 15 min/wash and then incubated with biotinylated secondary antibody (Cell and Tissue Staining Kit, R&D Systems, Inc) for 40 min. After that, the slides were washed three times in PBS for 15 min/wash and then incubated with high-sensitivity streptavidin-conjugated HRP (HSS-HRP) for 30 min. After two subsequent washes with PBS for 2 min/wash, the slides were incubated in the dark for 5 min with the DAB chromogen. Nuclear counterstaining was carried out using hematoxylin (Hematoxylin REF 05-06012/LBio-Optica, Milano, Italy). Finally, the slides were prepared for observation with coverslips, using a permanent mounting medium (Vecta Mount, Vector, H-5000). The observation of the sections was performed with an optical microscope (Leica DM 5000 B) connected to a digital camera (Leica DC 300F).

Two independent observers (F.C. and F.R.) examined the specimens on two separate occasions and performed a quantitative analysis to determine the percentage of cells positive for Hsp27, Hsp60, Hsp70, or Hsp90. All the observations were made at a magnification of 400× and the percentage of positive cells was calculated in a high-power field (HPF) and repeated for 10 HPF. The immunopositivity was expressed as the average percentage of all immuno-quantifications performed in each case for each Hsp. Statistical analyses were carried out using the GraphPad Prism 4.0 package (GraphPad Inc., San Diego, CA, USA). One-way ANOVA analysis of variance with Bonferroni post-hoc multiple comparison was used to find significant statistical differences. All data are presented as the mean ± SD, and the level of statistical significance was set at *p* ≤ 0.0001, as indicated in Figure 1B,C.

### 4.3. Exosomes Isolation and Characterization

Blood was collected and processed for plasma separation by centrifugation at 1800 × *g* for 30 min at 22 °C, as described before [7]. Then the plasma samples were stored at −80 °C until processing. Exosomes were isolated from plasma by several steps of differential ultracentrifugation and ultrafiltration. Briefly, 3 mL of plasma was centrifuged at 11,000 × *g* for 30 min to remove cell debris. The supernatant was diluted with PBS, then filtered through a 0.2 µm filter (Millex GP, Millipore, Darmstadt, Germany), followed by two-step ultracentrifugation at 110,000 × *g* for 2 h to pellet the exosomes. The exosomes were then washed in cold PBS and resuspended in 100 µL of PBS or 70 µL of RIPA (radioimmunoprecipitation assay) lysis buffer (0.3M NaCl, 0.1% SDS, 25 mm HEPES pH 7.5, 1.5 mm MgCl_2_, 0.2 mm EDTA, 1% Triton X-100, 0.5 mm DTT, 0.5% sodium deoxycholate) ( [7].

Exosomes were analyzed by DLS, TEM, and AFM in order to estimate size/diameter and morphology. WB was performed to detect the exosome markers Alix (mouse anti-Alix, 1A12 clone, Santa Cruz Biotechnology, Inc.) and CD81 (mouse anti-CD81, B-11 clone, Santa Cruz Biotechnology, Inc.), as well as to detect Hsp27, Hsp60, and Hsp90.

### 4.4. Dynamic Light Scattering (DLS)

DLS measurements were performed using a Brookhaven Instrument BI200-SM goniometer. The temperature was controlled to be 24 °C using a thermostatic recirculating bath. The time autocorrelation functions (TCF) were measured using a Brookhaven BI-9000 correlator and a 100 mW solid-state laser (Quantum-Ventus MPC 6000) tuned at λ = 532 nm. Measurements were taken at a 90° scattering angle. All samples were filtered through 0.2 μm cellulose acetate (Millex GP, Millipore) syringe filters to remove gross contaminants.

### 4.5. Transmission Electron Microscopy (TEM)

Exosomes obtained by ultracentrifugation were resuspended in PBS with the addition of 50 µL freshly made fixative (2.5% glutaraldehyde in PBS) for 30 min. After fixation, the preparations were mounted on formvar nickel grids by layering grids over 10 µL drops of exosome preparations for 10 min at 24 °C. Grid-mounted preparations were prepared for contrast staining by treating them with uranyl acetate (1%) for 5 min and with Reynolds’ solution for 5 min and, finally, rinsing them eight times in distilled water for 2 min. After this procedure, the grids were ready for electron microscopy (JEOL JEM 1220 TEM at 120 kV) [7].

### 4.6. Atomic Force Microscopy (AFM)

Exosomes obtained by ultracentrifugation were resuspended in PBS and 50 μL aliquots were deposited on freshly cleaved mica, then washed and dried under mild vacuum. Tapping mode AFM images were acquired in air using a multimode scanning probe microscope driven by a nanoscope V controller (Digital Instruments, Bruker, Kennewick, WA, USA). Single-beam uncoated silicon cantilevers (type SPM Probe Mikromasch) were used. The drive frequency was between 260 and 325 kHz; the scan rate was 0.25–0.7 Hz.

### 4.7. Western Blotting (WB)

WB was performed as described [7], using equal amounts of protein (50 µg) for each sample and horseradish peroxidase-conjugated sheep anti-mouse antibody (GE Healthcare Life Science, Milan, Italy). WB were detected using the Amersham enhanced chemiluminescence substrate (GE Healthcare Life Science) following the manufacturer’s instructions. Densitometric analyses of WB were performed using the National Institutes of Health Image J analysis program (version 1.40. National Institutes of Health, Bethesda, MD).

## Figures and Tables

**Figure 1 ijms-20-04496-f001:**
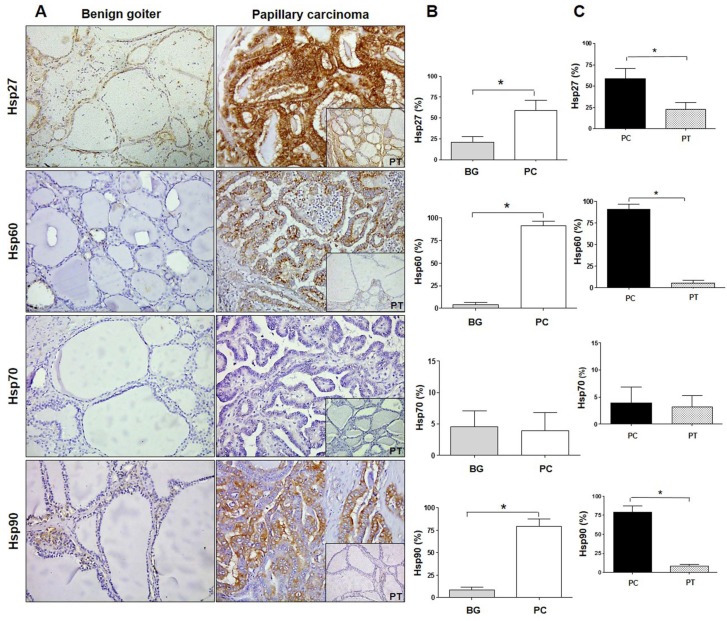
Immunohistochemistry for Hsps in benign goiter and papillary carcinoma. (**A**) Immunohistochemistry images of Hsp27, Hsp60, Hsp70, and Hsp90 in human thyroid tissue of benign (non-toxic) goiter and papillary carcinoma with pertinent normal peritumoral tissue (PT; insets at bottom right of each panel on the right). Magnification 200×. (**B**) Histograms showing the percentage of immunopositivity for Hsp27, Hsp60, Hsp70, and Hsp90 in benign goiter (BG) and papillary carcinoma (PC). Data are presented as the mean ± SD. * *p* ≤ 0.0001. (**C**) Histograms showing the percentage of immunopositivity for Hsp27, Hsp60, Hsp70, and Hsp90 in samples of papillary carcinoma (PC) and normal peritumoral tissue (PT). Data are presented as the mean ± SD. * *p* ≤ 0.0001.

**Figure 2 ijms-20-04496-f002:**
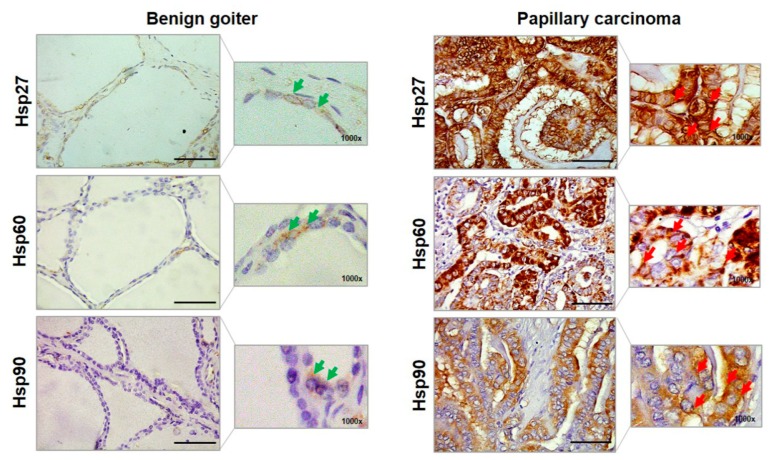
Representative images of the immunohistochemistry of benign goiter and papillary carcinoma for Hsp27, Hsp60, and Hsp90. Larger images were acquired at a magnification of 400× (scale bar: 100 µm); smaller images at 1000× allowed a better visualization of the cellular localization of immunopositivity. Green arrows, in benign goiter images, indicate for Hsp27 the cytosolic and perinuclear localizations; for Hsp60 the cytosolic and cytoplasmic granular (i.e., mitochondrial) localizations; and for Hsp 90 the cytosolic localization. Red arrows, in papillary carcinoma, indicate the cytoplasmic and plasma–cell membrane (or close to this membrane) localizations of Hsp27; the cytoplasmic diffuse, close to, and in plasma–cell membrane immunopositivity of Hsp60; and cytosolic and plasma cell–membrane localizations of Hsp90.

**Figure 3 ijms-20-04496-f003:**
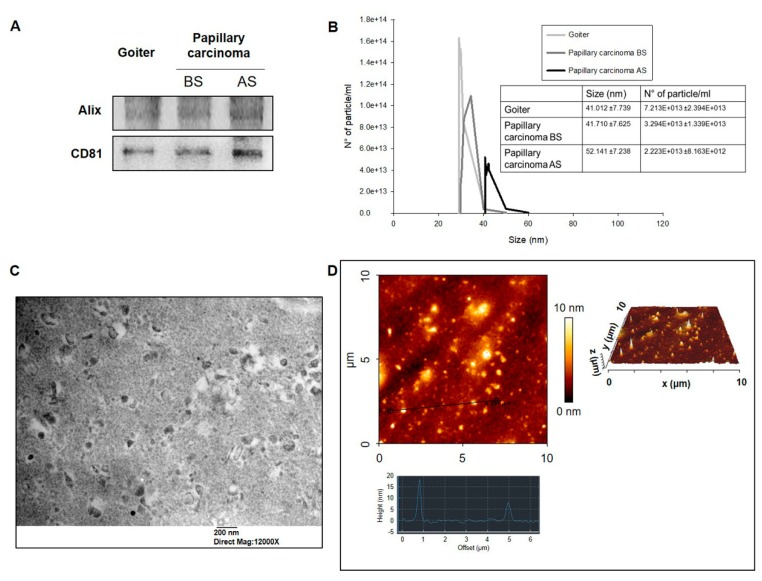
Exosome characterization. (**A**) Quantification and characterization by Western blotting (WB) of markers of exosomes from patients with goiter and papillary carcinoma before surgery (BS) and after surgery (AS). (**B**) Dynamic light scattering (DLS) characterization of exosomes showing their concentrations and their diameters. Mean ± SE of three different experiments are shown. (**C**) Transmission electron microscopy (TEM) (bar 200 nm) and (**D**) atomic force microscopy (AFM) images showing the typical characteristics of exosomes.

**Figure 4 ijms-20-04496-f004:**
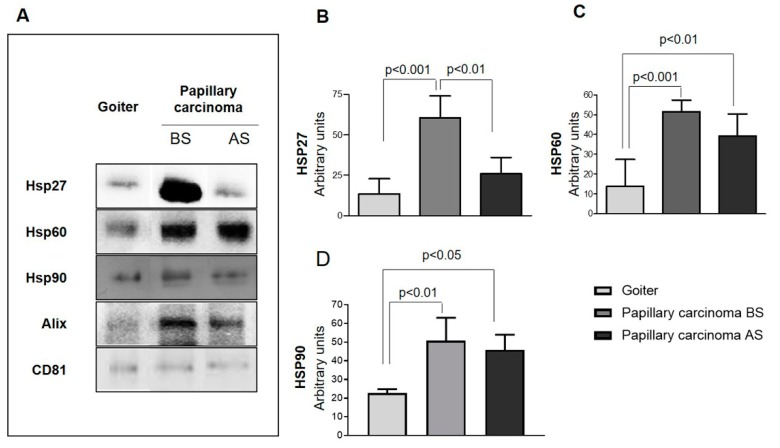
Western blotting data on molecular chaperones in exosomes. (**A**) Western blots showing the presence and levels of Hsp27, Hsp60, and Hsp90 in the exosomes studied. Visible are the higher levels of Hsp27, Hsp60, and Hsp90 in the exosomes from plasma from patients with papillary carcinoma before surgery (BS) compared with those obtained from either the same patients after surgery (AS) or from patients with goiter. The difference of Hsp27 levels between patients with papillary carcinoma BS and AS and with goiter was statistically significant (**B**), whereas Hsp60 (**C**) and Hsp90 (**D**) levels in patients with papillary carcinoma BS and AS were statistically significant when compared with patients with goiter.

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
