# Peer review of "Immunomorphological Pattern of Molecular Chaperones in Normal and Pathological Thyroid Tissues and Circulating Exosomes: Potential Use in Clinics"

_ijms, 2019, doi:10.3390/ijms20184496_

Round 1
Reviewer 1 Report
The authors have analyzed the expression of the different heat shock proteins in papillary thyroid cancers and compared it to benign goiters and normal peritumoral tissues. They found that Hsp27, Hsp 60 and Hsp 90 were increased in cancer tissues and in plasma exosome.
The manuscript is clearly written.
Major point:
It is important to complete the study by analyzing the expression levels of heat shock proteins also in the other subtypes of thyroid cancers: anaplastic, follicular, hurtle.
Author Response
Reviewer 1
English language and style are fine/minor spell check required
The authors have analyzed the expression of the different heat shock proteins in papillary thyroid cancers and compared it to benign goiters and normal peritumoral tissues. They found that Hsp27, Hsp 60 and Hsp 90 were increased in cancer tissues and in plasma exosome.
The manuscript is clearly written.
Major point:
It is important to complete the study by analyzing the expression levels of heat shock proteins also in the other subtypes of thyroid cancers: anaplastic, follicular, hurtle
Authors’ Reply: We thank the Reviewer for the positive comment on the contents of the manuscript and on the English language. We did check the manuscript for spelling errors and correct the few we found.
We appreciate the Reviewer’s suggestion about the possibility of studying the expression levels of Hsp27, Hsp60, Hsp70, and Hsp90 also in other subtypes of thyroid cancers. We think that those cancers may certainly be the subject of further study but are beyond the scope of the present work, which had, by necessity, to be designed to encompass a manageable range of pathologies, allowing a thorough and detailed analysis.
Reviewer 2 Report
Caruso Bavisotto C et al. reported that Hsp27, Hsp60 and Hsp90 show distinctive quantitative patterns in thyroid tissue 36 and circulating exosomes in PC as compared with BG.
It is unclear why authors chose these four chaperons. Why did they not choose Hspa5?? Authors performed immunohistochemical analysis to detect HSPs. Authors should perform real-time RT-PCR or ELISA to quantitate the levels of HSPs.
Author Response
Reviewer 2
Extensive editing of English language and style required
Caruso Bavisotto C et al. reported that Hsp27, Hsp60 and Hsp90 show distinctive quantitative patterns in thyroid tissue 36 and circulating exosomes in PC as compared with BG.
It is unclear why authors chose these four chaperons. Why did they not choose Hspa5??
Authors performed immunohistochemical analysis to detect HSPs. Authors should perform real-time RT-PCR or ELISA to quantitate the levels of HSPs.
Authors’ Reply: We thank the Reviewer for the comments. As requested by this Reviewer, the language was checked and edited by the authors and the spelling errors were corrected. In any case, this issue should not be of concern because one of the authors is an English speaker since birth and has extensive record of publications in top scientific journals, and another author is a specialist in languages with special training in scientific English writing and editing. In this regard, it is important to notice that Reviewer 1 stated that “English language and style are fine” and “The manuscript is clearly written.”
The Hsps taken into consideration are implicated in carcinogenesis in what concerns tumor-cell growth, multiplication, and dissemination. Consequently, the authors have chosen these Hsps on the basis of their own experience and on data in the literature, as reported in the introduction of the manuscript. A number of reports from both our and other groups demonstrate that these Hsps are also implicated in the pro-metastatic activity of the tumor microenvironment and are released into the circulation via exosomes. It has to be taken into account that the main objective of our work was to provide basic data to evaluate the possibility that these proteins could be useful biomarkers for diagnosis and follow-up in thyroid cancer. Also, given the data in the literature relating to these exosomal Hsps, we focused our attention on these proteins. Future research may focus on HSPA5, as suggested by this Reviewer, particularly in view of a very recent publication in which by bioinformatics this chaperone was found to be a possible candidate as biomarker in PC.
The objective and scope of our work were centered on a semi-quantitative investigation at the optical microscope level in tissue sections as dictated by the requirements of routine clinical pathology laboratory and by the type of specimens at our disposal. This kind of material is not amenable to ELISA, and PCR is beyond the scope of our work which, as said above, was focused on measuring the proteins considered as potential biomarkers with direct application in routine clinical pathology.
Round 2
Reviewer 1 Report
The authors have provided appropriate justifications for not performing the experiments requested.
Reviewer 2 Report
The manuscript has been improved.